# The Impact of Direct-Fed Microbials and Phytogenic Feed Additives on Prevalence and Transfer of Extended-Spectrum Beta-Lactamase Genes in Broiler Chicken

**DOI:** 10.3390/microorganisms8030322

**Published:** 2020-02-26

**Authors:** Eva-Maria Saliu, Hao Ren, Farshad Goodarzi Boroojeni, Jürgen Zentek, Wilfried Vahjen

**Affiliations:** Institute of Animal Nutrition, Freie Universität Berlin, Königin-Luise-Str. 49, 14195 Berlin, Germany; hao.ren@fu-berlin.de (H.R.); Farshad.Goodarzi@fu-berlin.de (F.G.B.); Juergen.Zentek@fu-berlin.de (J.Z.); Wilfried.Vahjen@fu-berlin.de (W.V.)

**Keywords:** extended-spectrum ß-lactamases, ESBL, phytobiotics, probiotics, essential oils, Lactobacillus, plasmid transfer, horizontal gene transfer, stress impact, conjugation

## Abstract

Poultry frequently account for the highest prevalence of extended-spectrum beta-lactamase (ESBL)-producing *Enterobacteriaceae* in livestock. To investigate the impact of direct-fed microbials (DFM) and phytobiotic feed additives on prevalence and conjugation of extended-spectrum beta-lactamase (ESBL)-producing *Enterobacteriaceae*, an animal trial was conducted. *Lactobacillus agilis* LA73 and *Lactobacillus salivarius* LS1 and two commercial phytogenic feed additives (consisting of carvacrol, cinnamaldehyde, and eugenol) were used as feed additives either alone or as a combination of DFM and phytogenic feed additive. An ESBL-producing *E.*
*coli* donor and a potentially pathogenic *Salmonella* Typhimurium recipient were inoculated at 5 × 10^9^ cells/mL in cecal contents from 2-week-old broilers. Conjugation frequencies were determined after 4 h aerobic co-incubation at 37 °C and corrected for the impact of the sample matrix on bacterial growth of donor and recipient. Surprisingly, indigenous *Enterobacteriaceae* acted as recipients instead of the anticipated *Salmonella* recipient. The observed increase in conjugation frequency was most obvious in the groups fed the combinations of DFM and phytogenic product, but merely up to 0.6 log units. Further, cecal samples were examined for ESBL-producing *Enterobacteriaceae* on five consecutive days in broilers aged 27–31 days. All samples derived from animals fed the experimental diet showed lower ESBL-prevalence than the control. It is concluded that *Lactobacillus* spp. and essential oils may help to reduce the prevalence of ESBL-harboring plasmids in broilers, while the effect on horizontal gene transfer is less obvious.

## 1. Introduction

Broiler chickens are the livestock with the highest prevalence of extended-spectrum beta-lactamase (ESBL)-producing *Enterobacteriaceae* in many regions of the world. In this context, *E. coli* and *Salmonella* spp. are the bacteria most commonly identified as the host of *bla* (ESBL encoding genes) carrying plasmids [1]. Transmission of ESBL-producing bacteria between animals happens rapidly and undetected, as no specific symptoms accompany the inoculation and establishment of ESBL-producing *Enterobacteriaceae* in broilers. These primarily non-pathogenic bacteria can, however, transfer mobile genetic elements to pathogenic bacteria and thereby cause infections, which are difficult to cure [2,3,4,5]. As they also may be transmitted to humans, these multi-resistant bacteria pose a major hazard to public health, causing tremendous costs worldwide [6,7,8,9].

As the problem gained global attention and the importance to intervene in the development and spread of antibiotic-resistant bacteria has gained high political priority, different measurements were developed to reduce the prevalence of ESBL-producing bacteria [8,10,11]. One possible approach to reduce the ESBL prevalence may be the use of feed additives, such as direct-fed microbials (DFM) or phytogenic feed additives [12,13]. The negative impact of DFMs, such as various *Lactobacillus* strains, on the prevalence of pathogens has been described frequently [12,13]. To our best knowledge, only three studies demonstrating the impact of DFMs and competitive exclusion cultures on prevalence and transfer of ESBL-producing *Enterobacteriaceae* in broilers have been published [14,15,16]. These studies comprise commercial products and competitive exclusion cultures, however no exact qualitative and quantitative specification of the containing microorganisms was provided.

Phytogenic products are used in poultry farming due to their beneficial impact on health and production [13,17]. These products can be grouped into four categories: herbs, botanicals, essential oils, and oleoresins [18]. Antibacterial activities against pathogens such as *Salmonella* spp. and *E. coli* among others have been observed in various essential oils [13,19,20]. Moreover, phytogenic feed additives were associated with a reduced plasmid transfer in *E. coli* [21].

Besides their prevalence, the spread of antibiotic resistance plays an important role in the dissemination of ESBL-producing *Enterobacteriaceae*. On the one hand, the transmission between animals and from animals to humans must be considered. Several studies have targeted this topic with one trial specifically covering the transmission of ESBL-producing *Enterobacteriaceae* between broilers receiving DFM [16]. On the other hand, horizontal gene transfer of ESBL-carrying plasmids must also be considered as antibiotic resistance genes are frequently exchanged between bacteria [22,23]. 

Thus, the approach of this study was to use DFMs with previously characterized components (qualitatively and quantitatively) as well as phytogenic products to reduce the prevalence and conjugation frequency of ESBL-producing *Enterobacteriaceae*. Results were obtained from in vivo (ESBL prevalence) and ex vivo (conjugation) experiments to investigate the natural occurrence and spread of ESBL genes as well as conjugation between artificially added strains.

## 2. Materials and Methods 

### 2.1. Animals and Husbandry

For the animal trial, newly hatched male Cobb 500 broiler chicks were randomly allocated to nine feeding groups with seven replicates each and reared for five weeks. Three animals were reared together in cages of 50 × 35 × 68 cm (depth × width × height) for two weeks, subsequently, the animal density was reduced to 1–2 animals per cage. Metal walls separating cages reduced the contact between animals. As the cage floor comprised a metal net, excreta were automatically excluded from the animals’ environment, reducing the contact of the animals with the excreta and thereby decreasing the risk for bacterial contamination. The initial temperature was 34 °C for 48 h and reduced by 3 °C weekly. After 72 h of constant light, a cycle of 18 h light and 6 h darkness was applied. Water and experimental diet were constantly available ad libitum. The nine experimental diets comprised a control diet (Table 1), two diets supplemented with *Lactobacillus salivarius* LS1 or *Lactobacillus agilis* LA73 (10^10^ cfu per kg feed) [24], two experimental diets were supplemented with the phytogenic products Formulation C or Formulation L (250 mg/kg feed; EW Nutrition, Germany) and four diets supplemented with a combination of one DFM and one phytogenic product (Table 2). Formulation C contained the essential oils carvacrol and cinnamaldehyde, while Formulation L additionally contained eugenol. These feed additives were chosen due to their ability to reduce the viability of the ESBL-producing *E. coli* strain ESBL10716 in a previous in vitro experiment [25]. The animal trial was approved by the Regional Office for Health and Social Affairs Berlin (LaGeSo Reg. A 0437/17).

### 2.2. Strains and Cultivation Conditions

The experimental design comprised the ESBL-producing donor strain *Escherichia coli* ESBL10682, derived from broiler excreta within the RESET program [26]. This strain belonged to the phylogenic group B1 and produced the enzyme CTX-M-1. Furthermore, the strain *Salmonella* Typhimurium L1219-R32 served as the recipient. Susceptibility of donor and recipient against various antibiotics was investigated by disc diffusion test (Appendix A Saliu et al., manuscript submitted). This conjugative pair was known to transfer plasmids in vitro at a conjugation frequency (CF) of 10^−4^–10^−5^ when incubated in Mueller Hinton 2 Broth (Sigma-Aldrich, Chemie GmbH, Germany) for 4 h [27]. All samples were cultivated aerobically at 37 °C for 4 h.

### 2.3. Collection of Samples

Pooled excreta samples were collected weekly from each experimental group. During days 13–17, these samples were derived from the cages used for the conjugation experiment only. The samples from the last collection originated from fewer cages as several animals had been sacrificed according to the trial design. At all remaining time points, all cages were sampled. After dilution with equal volumes of sterile glycerol (50% Glycerol, Carl Roth GmbH + Co. KG, Germany, 50% Phosphate Buffered Saline (PBS), Sigma-Aldrich Chemie GmbH, Germany), the samples were stored at −80 °C until further analyzed.

Two animals of each feeding group were sacrificed per day by cervical dislocation subsequent to anesthesia from day 13 to 17. The ceca were removed, immediately transferred to the laboratory facilities, and the content was collected. This procedure was repeated with one animal per feeding group and day on days 27–31. 

### 2.4. Conjugation Experiments

The donor and recipient strains were cultivated in Mueller Hinton 2 Broth supplemented with 8 µg/mL cefotaxime (CTX; Alfa Aesar, Thermo Fisher GmbH, Germany) or 300 µg/mL sulfamethoxazole/trimethoprim (SXT, Sigma-Aldrich, Chemie GmbH, Germany) respectively for 19 h. After washing the cells twice in PBS, cell concentrations were obtained photometrically and adjusted to 5×10^9^ cells/mL. Cecal samples were diluted 10-fold in a citric acid–Na-citrate buffer system (pH 6.2; Sigma-Aldrich, Chemie GmbH, Germany). Each 50 µL donor and recipient suspension were added to 900 µL diluted cecal samples in triplicates. The suspensions were thereafter incubated aerobically for 4 h and plated on selective MacConkey agar (Carl Roth GmbH + Co. KG, Germany) containing 300 µg SXT and 8 µg CTX per mL agar, to obtain transconjugants, 300 µg SXT/mL agar for recipient identification or 8 µg CTX/mL agar for the calculation of donor concentrations. CF was calculated as transconjugants/donor (CF/D) and transconjugants/recipients (CF/R). The negative control comprised cecal contents without donor or recipient and was plated on MacConkey agar containing 300 µg SXT and 8 µg CTX per mL agar after 4 h incubation.

### 2.5. Calculation of a Stress Impact Factor

As the sample matrix influences the growth of donor, recipient, and transconjugants, a bias arises when only evaluating CF based on donor or recipient cfu. Thus, the impact of stressors within the sample must be considered. A stress impact factor (SIF), correcting the results for this bias, aims to normalize the differences by incorporating control incubations [27]:(1)SIF=mean cfuctrmLmean cfustressmL

Here, cfu_ctr_ represents the cfu of the control and cfu_stress_ stands for the bacterial concentration (cfu) at a specific level of supplementation with a stress factor. No differences in growth are observed when the SIF equals 1, while a SIF smaller than 1 occurs when feed additives result in higher bacterial concentrations. On the other hand, a SIF larger than 1 outlines enhanced bacterial growth in the presence of the stressor compared to no supplementation. This factor can subsequently be used to calculate corrected bacterial concentrations (cfu_corr_), which most likely would be observed in the absence of the stressor by multiplication of the SIF and cfu_stress_: cfu_corr_= cfu_stress_×SIF(2)

SIF corrected CF were thereafter calculated as CF/D (SIF) = cfu_corr_(transconjugants)/ cfu_corr_(donor) and CF/R (SIF) = cfu_corr_(transconjugants)/ cfu_corr_(recipient). In the latter, the SIF for transconjugants equals the SIF for recipients. Hence, CF/R=CF/R (SIF) and CF/R (SIF) can be neglected.

### 2.6. Prevalence of ESBL-Producing Enterobacteriaceae

The excreta samples were thawed at room temperature and each 1 g sample was diluted in 4.5 mL Buffered Peptone Water (BPW, Carl Roth GmbH + Co. KG, Germany) as described previously [28]. The dilutions were plated on multiple MacConkey agar plates containing 2 µg CTX/mL agar as described previously [29,30,31] and incubated aerobically at 37 °C for 48 h. 

A similar procedure was applied for the cecal content samples collected on day 27–31. The cecal content was diluted in the double amount of PBS and directly spread on MacConkey agar containing 2 µg CTX/mL. The plates were evaluated after 48 h aerobic incubation at 37 °C. 

No pre-enrichment was conducted, as quantification of ESBL-producing *Enterobacteriaceae* was intended.

### 2.7. Statistics

The software IBM SPSS (Version 22, USA) was used for the statistical analysis of the results. The results are presented as mean values and standard deviation. To determine significance and subgroups, the non-parametric Kruskal–Wallis test and Mann–Whitney test were applied for CF and bacterial growth while the chi-squared test was applied for the prevalence of ESBL-producing *Enterobacteriaceae* in the cecal content. Differences at *p* < 0.05 were considered statistically significant and *p* values between 0.05 and 0.1 were accepted as trends. Pearson correlations were applied to identify correlations between short-chain fatty acid concentrations and CF and were considered significant at the 0.01 level (2-tailed).

## 3. Results

### 3.1. Donor and Recipient Growth

The different feed additives and their combinations did not show a significant impact on *Salmonella* Typhimurium L1219-R32 (*p* = 0.183) or *E. coli* ESBL10682 (*p* = 0.317) growth (Table 3). The initial concentration of approximately 8.7 log cfu/mL cecal content of each donor and recipient strain declined after 4 h of incubation to 7.5–7.9 log cfu/g cecal content for *E. coli* ESBL10682 and 7.9–8.4 log cfu *Salmonella* Typhimurium/g cecal content while the indigenous, SXT-resistant *Enterobacteriaceae* showed concentrations of 6.4–7.1 log cfu/g cecal content. On average, the *Salmonella* Typhimurium L1219-R32 showed slightly higher growth than the *E. coli* strain. The lowest growth was observed for SXT resistant *Enterobacteriaceae,* which also varied significantly (*p* = 0.002) between feeding groups (Table 3). Here, LS + L, LA + C, LA and Formulation C showed significantly lower numbers of SXT resistant *Enterobacteriaceae* of up to 0.72 log units than Formulation L or LA + L. However, these were not significantly lower than the observed amount of SXT resistant *Enterobacteriaceae* in samples derived from animals fed the control diet. 

### 3.2. Conjugation Experiments 

The donor/recipient pair was known from a previous in vitro study to show CF of 10^−5^–10^−4^ transconjugants/donor or transconjugants/recipient after 4 h of co-incubation [27]. In the aforementioned in vitro set up, a 10-fold lower initial concentration of donor and recipient was chosen compared to the setup of the present experiment. Still, in the present study, indigenous *Enterobacteriaceae* stepped in as plasmid acceptors instead of the intended *Salmonella* Typhimurium recipient at CF of 10^−5^–10^−4^ transconjugants/donor and 10^−3^ transconjugants/recipient (Table 4). Differences in CFD (SIF) were observed between different trial groups. The groups receiving feed supplemented with Formulation C, LS +C, LS + L or LA + L showed significantly higher CF/D (SIF) than the control group (Table 4). The applied feed additives did not affect CF/R. The negative control did not grow colonies on plates containing the combination of CTX and SXT or only CTX but on the plates containing SXT.

### 3.3. Prevalence of ESBL-Producing Enterobacteriaceae 

ESBL-producing *Enterobacteriaceae* (ESBL-PE) were detected in the excreta of newly hatched chicks and throughout the entire rearing period (Appendix A). To investigate the day-to-day differences of cecal ESBL-PE*,* the fifth week of the feeding trial was monitored closely. Significant differences (*p* = 0.001) were observed between feed groups both in regards to the number of days ESBL-PE were above the detection limit, as well as to the amount of detected ESBL-PE/g cecal content (Table 5). The groups LA and LS + L were negative at all sampling times. Quantitatively, groups LS + C, LA + L, LS, LA + C, Formulation C and Formulation L followed. When evaluating the qualitative results, LS + C and LA + L showed a lower prevalence than LS, LA + C, Formulation C and Formulation L. The ESBL-producing *Enterobacteriaceae* were not characterized further, but the microbial composition of the cecal content was described elsewhere [32].

## 4. Discussion

The threat that ESBL-producing *Enterobacteriaceae* from poultry pose to public health must be taken into serious consideration when discussing antibiotic resistance in farm animals. Different approaches to reduce the prevalence and spread of these bacteria in poultry by the use of feed additives showed promising results [14,15,16,21]. However, to our best knowledge, their impact on both prevalence and horizontal gene transfer has not been studied in poultry. A previous study by our working group focused on the identification of lactic acid bacteria and phytogenic products with inhibitory potentials against the ESBL-producing *E. coli* ESBL10716 [25]. Subsequently, the current trial followed to investigate their ability to reduce in vivo prevalence and ex vivo transfer of extended-spectrum beta-lactam resistance. 

To investigate the impact of these feed additives on conjugation frequency, donor and recipient strains were added to cecal contents obtained from 2-week-old broilers. The initial concentrations of the donor and recipient strains were chosen at very high levels compared to the common abundance of bacterial strains in the gastrointestinal tract of broilers [33,34,35] to investigate the theoretical possibility of gene transfer, as high concentrations increase the chance of detecting changes. Only a part of the initial donors and recipients added to the cecal contents were able to establish in the matrix, but *E. coli* and *Salmonella* Typhimurium concentrations still exceeded natural levels of single strains. Here, the detection of higher concentrations for *Salmonella* Typhimurium L1219-R32 than *E. coli* ESBL10682 was unexpected, as previous in vitro experiments with this mating pair constantly led to reverse results [27]. Apparently, the chosen habitat (cecum) was more favorable for the *Salmonella* strain. 

As both donor and recipient were able to survive the conditions in cecal contents, one would expect to find transconjugants after 4 h of co-incubation. Surprisingly, the complex system of an intestinal milieu revealed a tremendous impact on conjugation, as no *Salmonella* transconjugants could be recovered. The chosen *Salmonella*/*E. coli* mating pair has been established as an excellent system for in vitro experiments [27]. Furthermore, the growth of the *Salmonella* strain was also not affected by the cecal matrix. Therefore, other factors of the cecal matrix may have been present. During previous in vitro experiments, the simple addition of propionate for instance significantly reduced conjugation frequencies for the *Salmonella*/*E. coli* mating pair [27]. Cecal propionate concentrations were not significantly different between feed groups (Appendix A) [32] and no correlation between propionate (*p* = 0.662), acetate (*p* = 0.96) or total short-chain fatty acids (*p* = 0.905) and CF/D (SIF) were detected. Still, this and other factors of the cecal matrix (bacterial competition, quorum sensing molecules, intermediary metabolites) may have triggered a physiological state in either donor and/or recipient strain that inhibits the formation of the conjugation machinery [36,37,38]. 

Instead, within the period of 4 h, a time frame resembling the passage time of cecal content in 14-day-old broilers [39], *E. coli* donor successfully transferred the bla_CTX-M-1_ carrying plasmid to indigenous *Enterobaceriaceae*. This also leads to the conclusion that conjugation was not inhibited for the donor and thus, conjugation may have only been inhibited for the *Salmonella* recipient. As only the SXT resistant transconjugants were detected, one must expect that further, SXT sensitive enterobacteria additionally served as recipients and affected the CF observed in this experiment. As the negative control did not show growth on plates identifying donor or transconjugants, it was concluded that the chosen donor strain was accountable for the observed transconjugants. Further studies should investigate longer incubation times, simulating the establishment of donor and recipient in the gastrointestinal tract. Additionally, a wider screening of recipients should be included.

Comparing conjugation frequencies is a common method applied when evaluating horizontal gene transfer [40]. To calculate the CF, the amount of transconjugants is either divided by the number of recipients or donors. Results from the present study display major differences between these approaches and stress the importance of considering both methods when evaluating plasmid transfer. Also, the impact of the environment on bacterial growth is frequently neglected. Thereby, a bias arises [40,41,42], which was circumvented in the current study by applying the stress impact factor (SIF). In the present study, the results of SIF correction were rather similar to the CF/D and CF/R values besides group LA, Formulation C and Formulation L, where differences of 0.5, 0.4, and 0.3 log cfu/mL were observed, respectively. In these groups, the SIF for transconjugant and recipient growth were among the most pronounced at 2.8, 2.3, and 0.4 respectively. This highlights the importance of considering changes in bacterial growth when interpreting results from conjugation trials. Observed CF/R in this ex vivo setup exceeded results from previous in vitro experiments, while the CF/D was rather similar to earlier findings [27]. This might be explained by the amount of different possible indigenous recipients compared to one *Salmonella* recipient strain. 

The lowest detected CF/D (SIF) among indigenous enterobacteria was observed when fed the experimental diets supplemented with *L. agilis* or Formulation L. Similarly, it was previously described that *Lactobacillus plantarum* strains can reduce CF/R in vitro, independent of their ability to produce bacteriocins [43]. Similar results were observed in an in vitro trial with *Klebsiella pneumoniae* (SHV-5) and *Salmonella enterica* serovar Typhimurium (CTX-M-15) donors and an *Escherichia coli* K-12 recipient in the presence of different *Bifidobacterium* spp., reducing transconjugant counts with up to 2.6 logs [44]. These results were confirmed in in vivo trials with gnotobiotic mice, where a strain and incubation time-dependent reduction of transconjugants of up to 3.3 logs was reported. However, no information is provided on the impact of DFM on horizontal gene transfer of *bla* in a complex matrix comprising a diverse microbial community. Compared to the mentioned studies, the reduction of CF observed in the current study in the presence of *L. agilis* was less pronounced and statistically not significantly different from the control. The results from the current study also revealed species-specific differences as, opposite to *L. agilis*, *L. salivarius* induced a slight increase of CF/D, CF/R, and CF/D (SIF) compared to the control. 

The phytobiotic product Formulation C seemed to reduce CF/D, but correction for stress impact on bacterial growth reversed this into an increase. On the contrary, the observed increase in CF/D by the group receiving Formulation L was reversed into a decrease when corrected by SIF. Similarly, components of *Thymus vulgaris* essential oils reduced the transfer of the *bla*-carrying pKM101 plasmid between *E. coli* strains. The highest reduction was observed as an effect of linalool supplementation, followed by S- and R-carvone, eugenol, and borneol [21]. This implies that different phytogenic compounds affect CF differently. 

Interestingly, the combination of DFM and phytobiotic feed additives had an enhancing impact on conjugation frequencies. The only exception was observed for the combination of *L. agilis* and Formulation L, where a minor numeric reduction was observed. In summary, a non-significant reductive impact on conjugation (CF/D (SIF)) was observed for *L. agilis* and Formulation L, while a significantly higher CF was detected in Formulation C, LS + C, LS + L, and LA + C than in the control. This suggests that the chosen DFM and phytobiotics are unlikely to reduce conjugation frequencies of the investigated ESBL-carrying plasmid in broilers. 

In poultry, ESBL-PE prevalence of up to 100% were previously reported [1]. This corresponds with findings from excreta samples at week 4, where all samples were tested positive (Appendix A). The early detection of ESBL-PE in newly hatched broiler chicks corresponds with results from the literature, identifying chicks and eggs as a potential risk factor for transmission between farms [45,46]. In cecal contents, two feed groups were negative for ESBL-PE at all sampling times. Comparing quantitative and qualitative results from the other feed groups, differences appear. The groups characterized as equal in the qualitative approach (20% prevalence) showed obvious quantitative differences. Qualitative evaluation of ESBL-PE prevalence, often after pre-enrichment, is commonly used to report antimicrobial resistance in poultry [1]. The results of this study suggest that qualitative evaluations may distort the picture. 

It was previously shown in vivo*,* that DFM can reduce the prevalence of ESBL-producing *E. coli* in the ceca of broiler chicks [14,15], corresponding with the results from this in vivo trial. Another study investigated the transmission of ESBL-producing *E. coli* (CTX-M-1) between animals, where animals perceiving the DFMs were less susceptible to ESBL-producing *E. coli* and excreted lower numbers of these bacteria [16]. The effect observed in these studies were based on competitive exclusion. In contrast to the present study, the aforementioned trials performed an ESBL-PE challenge. Additionally, commercial products comprising a diverse bacterial community, which was not characterized quantitatively, was used compared to the single-strain approach of the present study.

Besides the impact of the DFM, phytogenic feed additives may have an effect on the prevalence of ESBL-PE [47,48]. A reduction of ESBL-PE prevalence was observed both qualitatively and quantitatively in the presence of the phytogenic products in cecal contents. As quantitative differences were observed between the products, the most severe effect was observed when combined with a DFM strain. 

The results of this study displayed differences in the efficiency of different lactobacilli strains and phytobiotic products regarding the ability to reduce ESBL-PE prevalence and their plasmid transfer. In addition, no consistent correlation between the ability to reduce the prevalence and the CF was observed. This suggests that feed additives reducing the ESBL-PE prevalence should be combined with supplements targeting plasmid transfer to achieve the highest possible reduction of ESBL-producing bacteria. As this may sound logical in theory, the interaction of combinations may lead to different results [32]. Thus, it is crucial to determine the combined effect of such products, as they may be additive but also can neutralize or even reverse the effect of the single components. In this study, only one combination (LA + L) did not increase CF. With regards to the ESBL-PE prevalence, 2 (LS + C, LS) of 4 combinations were superior to the quantitative results induced by the DFM supplementation and all 4 combinations performed better than the single supplementation of phytobiotics. Further studies, where the broilers are challenged with defined amounts of ESBL-producing *E. coli,* should be conducted in the future to compare the results with these results from natural colonization with ESBL-PE. 

## 5. Conclusions

Out of the tested feed additives, the effect of DFM on ESBL-PE prevalence was superior to the phytobiotic products. *L. agilis* showed the most promising ability to reduce both the prevalence of ESBL-producing *Enterobacteriaceae* as well as the ESBL-carrying plasmid transfer between *Enterobacteriaceae*. Combinations of phytogenic additives and DFM did not enhance the effect of the single components on CF. The impact of DFM and phytobiotic feed additives on conjugation was less obvious than the impact on ESBL-PE prevalence.

## Figures and Tables

**Table 1 microorganisms-08-00322-t001:** Composition of the basal diet.

Ingredients	g/kg
Maize	320.3
Wheat	247.8
Soybean meal 49% CP	323.3
Soybean oil	59.5
Mineral–Vitamin Premix ^1)^	12.0
Limestone	14.6
Monocalcium phosphate	18.4
Salt	1.0
DL-Methionine	1.8
L-Lysine	1.3
Skim milk powder	0.3
**Nutrient Composition**	
Crude Protein (%)	22.00
Crude Fat (%)	8.19
Crude Fiber (%)	2.42
Methionine (%)	0.51
Lysine (%)	1.28
Threonine (%)	0.84
Calcium (%)	0.96
Phosphorus (%)	0.80
**Calculated Apparent Metabolizable Energy**	
AME_N_ (MJ/kg) ^2)^	12.6

**Table 2 microorganisms-08-00322-t002:** Feed additives applied to feed groups.

Feed Group	Diet Supplementation
Control	None
LS	DFM 1:10^10^ cfu *Lactobacillus salivarius* LS1/kg diet
LA	DFM 2: 10^10^ cfu *Lactobacillus agilis* LA73/kg diet
Formulation C	Phytogenic product 1: 0.25 g Formulation C/kg diet
Formulation L	Phytogenic product 2: 0.25 g Formulation L/kg diet
LS + C	0.25 g Formulation C + 10^10^ cfu *Lactobacillus salivarius* LS1/kg diet
LS + L	0.25 g Formulation L + 10^10^ cfu *Lactobacillus salivarius* LS1/kg diet
LA + C	0.25 g Formulation C + 10^10^ cfu *Lactobacillus agilis* LA73/kg diet
LA + L	0.25 g Formulation L + 10^10^ cfu *Lactobacillus agilis* LA73/kg diet

**Table 3 microorganisms-08-00322-t003:** Impact of direct-fed microbials (DFM) and phytobiotic feed additives on bacterial growth after 4 h incubation of inoculated cecal samples with the donor (*E. coli* ESBL10682) and recipient (*Salmonella* Typhimurium L1219-R32) strains [log_10_(cfu/mL)].

Trial Group	*E. coli* ESBL10682	*Salmonella* Typhimurium L1219-R32	Indigenous, SXT Resistant *Enterobacteriaceae* ^2^
Control	6.86 ± 0.48	7.10 ± 0.37	5.93 ± 0.61 ^abe^
LS	6.81 ± 0.27	7.04 ± 0.39	5.92 ± 0.52 ^be^
LA	6.90 ± 0.45	6.92 ± 0.47	5.49 ± 0.42 ^abcd^
Formulation C	6.94 ± 0.31	7.06 ± 0.42	5.50 ± 0.52 ^abcd^
Formulation L	6.91 ± 0.28	7.36 ± 0.24	6.12 ± 0.80 ^e^
LS + C	6.76 ± 0.54	7.21 ± 0.31	5.85 ± 0.49 ^abde^
LS + L	6.56 ± 0.71	7.14 ± 0.32	5.40 ± 0.64 ^abcd^
LA + C	6.49 ± 0.73	7.10 ± 0.37	5.45 ± 0.68 ^abcd^
LA + L	6.59 ± 0.53	7.14 ± 0.24	6.02 ± 0.51 ^e^
p ^1^	0.317	0.183	0.002

^1^ Significant differences were determined using the Kruskal–Wallis test, ^2^ different letters indicate significant differences (*p* < 0.05, Mann–Whitney test) in the abundance of indigenous, SXT resistant *Enterobacteriaceae* after 4 h incubation between trial groups.

**Table 4 microorganisms-08-00322-t004:** Impact of DFM and phytobiotic feed additives on CF after 4 h incubation of donor (*E. coli* ESBL10682) and the recipient (*Salmonella* Typhimurium L1219-R32) in cecal contents [log_10_(CF ^1^)].

Trial Group	CF/D ^3^	CF/R	CF/D (SIF) ^3^
Control	−4.43 ± 0.45 ^ab^	−3.46 ± 1.01	−4.43 ± 0.45 ^a^
LS	−4.21 ± 0.57 ^b^	−3.32 ± 0.74	−4.29 ± 0.57 ^ab^
LA	−5.07 ± 1.46 ^a^	−3.65 ± 1.12	−4.57 ± 1.46 ^a^
Formulation C	−4.60 ± 0.40 ^a^	−3.16 ± 0.68	−4.20 ± 0.40 ^b^
Formulation L	−4.27 ± 0.52 ^ab^	−3.48 ± 0.96	−4.66 ± 0.52 ^a^
LS + C	−4.10 ± 0.52 ^b^	−3.19 ± 0.56	−3.98 ± 0.52 ^b^
LS + L	−3.98 ± 0.96 ^b^	−2.82 ± 0.71	−3.82 ± 0.96 ^b^
LA + C	−3.95 ± 0.55 ^b^	−2.90 ± 0.68	−3.98 ± 0.55 ^b^
LA + L	−4.10 ± 0.94 ^b^	−3.53 ± 0.86	−4.33 ± 0.94 ^ab^
p ^2^	0.010	0.172	0.031

^1^ CF = conjugation frequency; ^2^ significant differences were determined using the Kruskal–Wallis test, ^3^ different letters indicate significant differences (*p* < 0.05, Mann–Whitney test) in conjugation frequency between trial groups within a column; CF/D = transconjugants/donor; CF/R = transconjugants/recipients; SIF = stress impact factor.

**Table 5 microorganisms-08-00322-t005:** Impact of probiotic and phytobiotic feed additives on the prevalence of ESBL-producing *Enterobacteriaceae* in cecal contents of broilers [cfu/g].

Day	27	28	29	30	31	Prevalence per Trial Group (%)
Trial Group						
Control	13	nd	7	8340	nd	60 ^a^
LS	20	nd	nd	nd	40	40 ^b^
LA	nd	nd	nd	nd	nd	0 ^b^
Formulation C	13	nd	nd	nd	80	40 ^b^
Formulation L	7	nd	nd	539	nd	40 ^b^
LS + C	nd	nd	20	nd	nd	20 ^b^
LS + L	nd	nd	nd	nd	nd	0 ^b^
LA + C	nd	63	nd	25	nd	40 ^b^
LA + L	nd	nd	nd	7	nd	20 ^b^
Prevalence (%): positive samples per day	44	11	22	44	22	
p **^1^**						0.001

nd: not detected. ^1^ Significant differences were determined using the chi-squared test, different letters indicate significant differences (*p* < 0.05) in the overall prevalence of ESBL-producing *Enterobacteriaceae* between trial groups.

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
