# Peer review of "The Impact of Direct-Fed Microbials and Phytogenic Feed Additives on Prevalence and Transfer of Extended-Spectrum Beta-Lactamase Genes in Broiler Chicken"

_microorganisms, 2020, doi:10.3390/microorganisms8030322_

Round 1

Reviewer 1 Report

There are many conclusions based on assumptions with no scientific data to support the assumptions.  Although conjugation between Enterobacteriaceae occurs in liquid this depends on the length of pili. The preferred way is via solid support on filters.

The donor and recipient have not been phenotypically or genotypically characterized in these experiments or the previous paper that is referenced. At a minimum susceptibility testing should have been done. Additionally Incompatibility typing of the plasmids present in the donor, recipient and transconjugants should have been done. Indole testing could also have distinguished Salmonella and E. coli.

The biggest problem in my opinion is that ESBLs were present in the normal flora of the birds throughout the experiment. That would be such a problem if through characterization were done on the donor and recipient. The authors say the normal flora acted as recipients. That has been seen before in poultry. Here the authors only seem to base that assumption on enumeration data. IncN is one of the most common incompatibility plasmids that possess CTX-M genes.

Author Response

Dear editor

We appreciate the comments and advice and have improved the manuscript according to the reviewers suggestions as follows. The changes are tracked in the manuscript and are summarized below. We hope the amended version will now fulfil the requirements

Kind regards, Eva Maria Saliu

Reviewer 1

Thank you for your comments and suggestions.

COMMENT

There are many conclusions based on assumptions with no scientific data to support the assumptions.  Although conjugation between Enterobacteriaceae occurs in liquid this depends on the length of pili. The preferred way is via solid support on filters.

ANSWER

The liquid set up was chosen for the conjugation experiments to mimic the conditions of the gastrointestinal tract. It was known from previous experiments that the intended conjugation pair performs well in liquid under various conditions, as stated in lines 110 and 199 and published in Saliu, E.M.; Eitinger, M.; Zentek, J.; Vahjen, W. Nutrition Related Stress Factors Reduce the Transfer of Extended-Spectrum Beta-Lactamase Resistance Genes between an Escherichia coli Donor and a Salmonella Typhimurium Recipient In Vitro. Biomolecules 2019, 9, doi:10.3390/biom9080324. Previous experiments performed by other work groups also used the liquid method to determine changes in conjugation frequency. We therefore followed those published assays in general. Here are some examples:

Händel, N.; Otte, S.; Jonker, M.; Brul, S.; ter Kuile, B.H. Factors That Affect Transfer of the IncI1 beta-Lactam Resistance Plasmid pESBL-283 between E-coli Strains. PLoS ONE 2015, 10, e0123039.

Franiczek, R.; Dolna, I.; Krzyzanowska, B.; Szufnarowski, K.; Kowalska-Krochmal, B. Conjugative transfer of multiresistance plasmids from ESBL-positive Escherichia coli and Klebsiella spclinical isolates to Escherichia coli strain K12C600. Adv. Clin. Exp. Med. 2007, 16, 239–247

Smet, A.; Rasschaert, G.; Martel, A.; Persoons, D.; Dewulf, J.; Butaye, P.; Catry, B.; Haesebrouck, F.; Herman, L.; Heyndrickx, M. In situ ESBL conjugation from avian to human Escherichia coli during cefotaxime administration. J. Appl. Microbiol. 2011, 110, 541–549.

COMMENT

The donor and recipient have not been phenotypically or genotypically characterized in these experiments or the previous paper that is referenced. At a minimum susceptibility testing should have been done. Additionally Incompatibility typing of the plasmids present in the donor, recipient and transconjugants should have been done. Indole testing could also have distinguished Salmonella and E. coli.

ANSWER

The intended donor and recipient were characterized previously. As stated in lines 107-109 the donors phylogenic group is B1 and it carries the genes for CTX-M-1. Susceptibility testing was performed for both donor and intended recipient. As the paper where this information is provided is not published yet, we added this information to the supplementary data: Lines 109-110: ‘Susceptibility of donor and recipient against various antibiotics was investigated by disc diffusion test (Table S1, Saliu et al., manuscript submitted)’.

No indigenous bacteria were able to grow on the plates containing 8 µg CTX/mL agar, it is therefore very unlikely that indigenous bacteria acted as donors for the detected transconjugants.

COMMENT

The biggest problem in my opinion is that ESBLs were present in the normal flora of the birds throughout the experiment. That would be such a problem if through characterization were done on the donor and recipient.

ANSWER

High concentrations of CTX and SXT were used to identify donor and recipients and the cecal content was tested for bacteria growing under these conditions (negative control, described in lines 138-139). We could not identify any bacteria beside the chosen donor strain (and transconjugants) that were able to grow on agar plates containing 8 mg CTX/mL agar. On the agar plates containing SXT, both the anticipated recipient (Salmonella) and indigenous Enterobacteria were observed.

We clarified this in lines 205-206: ‘The negative control did not grow colonies on plates containing the combination of CTX and SXT or only CTX, but on the plates containing SXT.’

And in lines 271-273: ‘As the negative control did not show growth on plates identifying donor or transconjugants, it was concluded that the chosen donor strain was accountable for the observed transconjugants.’

COMMENT

The authors say the normal flora acted as recipients. That has been seen before in poultry. Here the authors only seem to base that assumption on enumeration data. IncN is one of the most common incompatibility plasmids that possess CTX-M genes.

ANSWER

We agree that indigenous bacteria acting as recipients in the intestinal content of broilers was reported before, which is not surprising. What was surprising is the fact that the well-established in vitro model did not perform conjugations when other recipients were available.

Still, this is just an observation that was made and the main focus of our work was the impact of feed additives on conjugation (this was not investigated previously) and prevalence (different publications on this topic are cited (Nuotio et al., 2013, Ceccarelli et al., 2017, Methner et al., 2019).

Further changes:

Line 36: ‘spp’ not italic

Line 170: ‘p’ not italic

Lines 216, 261, 365: ‘Table S1’ changed to ‘Table S2’

Line 292: ‘the’ was deleted

Lines 320, 365: ‘Table S2’ changed to ‘Table S3’

Line 366: ‘Table S3’ changed to ‘Table S4’

Reordering references:

Lines 245, 478: ‘32’ changed to ‘33’

Lines 245, 481: ‘33’ changed to ‘34’

Lines 245, 484: ‘34’ changed to ‘35’

Lines 261, 349, 475: ‘35’ changed to ‘32’

Reviewer 2 Report

Saliu et al. present results of their experimental study working out the impact of direct-fed microbials and phytogenic feed activities on ESBL-prevalence and gene transfer in broiler chicken. The study is well written and of high scientific impact in the one-health approach, reducing ESBL-prevalence in animals and humans.

minor remarks:

1) It is difficult to understand, what the superscript letters in the tables indicate. Please clarify that in more detail.

2) Please state, which ESBL-producing species were detected in cecal contents of broilers.

Author Response

Dear editor

We appreciate the comments and advice and have improved the manuscript according to the reviewers suggestions as follows. The changes are tracked in the manuscript and are summarized below. We hope the amended version will now fulfil the requirements

Kind regards, Eva Maria Saliu

Reviewer 2

Comments and Suggestions for Authors

Saliu et al. present results of their experimental study working out the impact of direct-fed microbials and phytogenic feed activities on ESBL-prevalence and gene transfer in broiler chicken. The study is well written and of high scientific impact in the one-health approach, reducing ESBL-prevalence in animals and humans.

minor remarks:

Thank you for your kind remarks and valuable suggestions.

COMMENT 1) It is difficult to understand, what the superscript letters in the tables indicate. Please clarify that in more detail.

ANSWER 1) The different letters indicate significant differences in bacterial abundance, conjugation frequency or prevalence as calculated by pairwise comparison.

We revised the manuscript as follows:

Line 191-193: ‘2different letters indicate significant differences (p<0.05) between groups within a column (Mann-Whitney test)’ was revised: ‘2different letters indicate significant differences (p<0.05, Mann-Whitney test) in the abundance of indigenous, SXT resistant Enterobacteriaceae after 4 h incubation between trial groups’

Lines 210-211: ‘3different letters indicate significant differences (p<0.05) between groups within a column (Mann-Whitney test)’ was revised: ‘3different letters indicate significant differences (p<0.05, Mann-Whitney test) in conjugation frequency between trial groups within a column’

Lines 227-228: ‘1 Significant differences were determined using the chi-squared test, different letters indicate significant differences (p<0.05) in overall prevalence of ESBL-producing Enterobacteriaceae between trial groups’

COMMENT 2) Please state, which ESBL-producing species were detected in cecal contents of broilers.

ANSWER 2) Thank you for your comment. The individual species of the indigenous ESBL-producing Enterobacteriaceae was not investigated further, as it was not within the scope of the study. However, it is true that this would be interesting and we will keep this in mind as a possible analyze for future trials. The microbial composition in the ceca was described in the publication Ren, H.; Vahjen, W.; Dadi, T.; Saliu, E.-M.; Boroojeni, F.G.; Zentek, J. Synergistic Effects of Probiotics and Phytobiotics on the Intestinal Microbiota in Young Broiler Chicken. Microorganisms 2019, 7, doi:10.3390/microorganisms7120684.

We added this information to lines 220-221: The ESBL-producing Enterobacteriaceae were not characterized further, but the microbial composition of the cecal content was described elsewhere [32].

Further changes:

Line 36: ‘spp’ not italic

Line 170: ‘p’ not italic

Lines 216, 261, 365: ‘Table S1’ changed to ‘Table S2’

Line 292: ‘the’ was deleted

Lines 320, 365: ‘Table S2’ changed to ‘Table S3’

Line 366: ‘Table S3’ changed to ‘Table S4’

Reordering references:

Lines 245, 478: ‘32’ changed to ‘33’

Lines 245, 481: ‘33’ changed to ‘34’

Lines 245, 484: ‘34’ changed to ‘35’

Lines 261, 349, 475: ‘35’ changed to ‘32’